# Basal Reactivity Evaluated by Infrared Thermography in the “Caballo de Deporte Español” Horse Breed According to Its Coat Color

**DOI:** 10.3390/ani12192515

**Published:** 2022-09-21

**Authors:** Ester Bartolomé, Davinia I. Perdomo-González, María Ripollés-Lobo, Mercedes Valera

**Affiliations:** Department of Agronomy, Escuela Técnica Superior de Ingeniería Agronómica (ETSIA), University of Seville, Utrera Rd. Km 1, 41013 Seville, Spain

**Keywords:** eye temperature, genetic breed group, heritability, genetic parameters, genetic groups, residual variance heterogeneity, show jumping competitions

## Abstract

**Simple Summary:**

The importance of the coat color in horses has been present since their domestication. The interactive effect of coat color-determinant genes on other traits, such as behavior, makes it a factor of importance in horse selection, as it determines the way the horse perceives and reacts to its environment. Eye temperature at rest (ETR) assessed with infrared thermography was used as a suitable non-invasive tool to assess basal reactivity. The aim of this study was to evaluate the influence of coat color on basal reactivity assessed with infrared thermography, determine their relationship with the results obtained in show jumping competitions, and to estimate the genetic parameters for ETR to test its suitability for genetic selection. Our results indicated differences in ETR due to six different factors. We also discovered a positive correlation with ranking in show jumping, indicating that less reactive horses were more likely to achieve better rankings. The genetic results indicated that part of the variability present in the basal reactivity due to coat color has an environmental origin. The heritability values indicate that ETR has a suitable genetic basis to be used in the breeding program to select for basal reactivity due to coat color.

**Abstract:**

Horses have been valued for their diversity of coat color since prehistoric times. In particular, the pleiotropic effect that coat color genes have on behavior determines the way the horse perceives and reacts to its environment. The primary aim of this study was to evaluate the influence of coat color on basal reactivity assessed with infrared thermography as eye temperature at rest (ETR), determine their relation with the results obtained by these horses in Show Jumping competitions and to estimate the genetic parameters for this variable to test its suitability for genetic selection. A General Linear Model (GLM) and Duncan post-hoc analysis indicated differences in ETR due to coat color, sex, age, location, and breed-group factors. A Spearman’s rank correlation of 0.11 (*p* < 0.05) was found with ranking, indicating that less reactive horses were more likely to achieve better rankings. Heritability values ranged from 0.17 to 0.22 and were computed with a model with genetic groups and a model with residual variance heterogeneity. Breeding values were higher with the last genetic model, thus demonstrating the pleiotropic effect of coat color. These results indicate that ETR has a suitable genetic basis to be used in the breeding program to select for basal reactivity due to coat color.

## 1. Introduction

The importance of coat color diversity in horses dates to prehistoric times [1]. The relevance of coat coloration in research into horse domestication and selection has been previously demonstrated [2,3,4]. However, in addition to the importance of the coat color in defining the standard of a breed or a horse’s sale price, the pleiotropic effect that the genes which determine the coat color have on diseases as the effect of MC1R gen on vitiligo or melanoma [5,6], or the effect of roan gen on lethal diseases [7,8]; conformation as congenital ocular abnormalities linked to different coat colors [9,10]; functionality, as gaits [11], performance in race, and endurance horses [12,13]; and behavior, as the influence of MC1R and ASIP genes on boldness [14], the silver gene on fear reactions [15], and influence on habituation, temperament, and stress of different coat colors [16,17,18,19,20], makes it a factor of paramount importance in horse selection and improvement. Particularly, different temperaments in horses have been traditionally related to phaneroptic and genetic characteristics, such as coat color, which determine the way the horse perceives their environment and reacts towards it. For instance, gray and bay horses have been associated with calm, noble temperaments [14,15], whereas chestnut and black horses have been related with more nervous, unpredictable temperaments [14,16].

Reactivity and stress induce behavioral and physiological changes on the animal. Behavioral changes will vary on intensity and nature due to the stressor considered [21,22]. Whereas the physiological changes have been monitored by measuring heart rate (HR) response, blood lactate concentrations, oxygen uptake in relation to exercise intensity [23,24], salivary, hair, and blood cortisol [25,26,27,28], or immune cell proliferation [29,30,31]. However, these are all relatively difficult measurements to take, as these methods either require laboratory conditions (oxygen uptake, salivary, and hair cortisol), a blood sample (lactate, blood cortisol, and immune cells) or touching the horse during the assessment (heart rate), thus biasing the results.

Recently, a non-invasive tool to measure physiological changes, eye temperature assessed with infrared thermography (IRT) technology, has demonstrated great potential for assessing physiological stress in horses and also correlates with traditional, invasive physiological measures such as lactate concentration in blood [32,33,34]. This tool can detect minor changes in temperature by measuring the radiated electromagnetic energy produced by the horse as a physiological response to exercise. Thus, the increase of catecholamine concentrations, in addition to blood flow responses produced during a reaction to a potential threaten stimulus, will lead to changes in heat production and heat loss in small areas around the medial posterior palpebral border of the lower eyelid and the lacrimal caruncle, since both contain rich capillary beds innervated by the sympathetic system [35].

As occurs with heart rate or lactate levels, eye temperature assessed with IRT is not only dependent on the aerobic capacity of the horse [33,34], but also on inherited parameters such as breed [36,37] or age [33,36,38]. This issue is of increased importance in crossbreeds, as the “Caballo de Deporte Español” (CDE) horse. The CDE is a recent composite breed formed from crosses with other horse breeds. It was founded in 2002 and the main goal of its breeding program is to obtain a horse with good functional conformation, temperament, and health, which can perform well in national or international sports events [39]. The coat color of the CDE horse is highly diverse, since this breed was formed by crossing a wide range of horse breeds with different origins and morphological types.

The aims of this study were, first, to test the influence of coat color and different environmental effects (age, sex, location, and breed origin) on basal reactivity in CDE before a performance test, using IRT; second, to assess the relation between this measurement and the horses’ results in show jumping competitions; and third, to estimate the genetic parameters and breeding values for this variable, to test its suitability for genetic selection.

## 2. Materials and Methods

### 2.1. Animals

Measurements were taken from 471 animals (311 stallions and 160 mares) of the Caballo de Deporte Español (CDE) breed, aged from 4 to 18 years old. From this group of animals, a pedigree matrix of 8245 animals was built (see Section 2.4 for further description).

The animals measured were selected according to their participation in the official show jumping competition which was held the day after the study. All the owners had been contacted and informed about the study using their animals. Only the animals whose owners agreed to participate in the study were used for the analyses. The owners were asked to bring their horses to the equestrian center two days before the official competition in order to participate in this study. All the experiments were performed in accordance with the guidelines in EU Directive 2010/63/EU.

Due to the composite nature of the CDE, many different breeds were included in the studbook as relatives. In order to consider the influence of the genetic composition on the reactivity related to coat color of the CDE studied, five breed groups were assessed, according to their origin. When the majority of an animal’s ancestors came from a breed in one of these groups, it was classified as belonging to that group [39,40]. The largest breed group was BG4 or the PRE breed group, with 43.3% (204 horses) of the CDE horses belonging to it, while the smallest was the BG2 or the Netherlands breed group, containing 9.6% (45 horses) of the CDE horses (Table 1).

As regards coat color information, the pedigree animals measured were grouped according to coat color classification [41], with 78 (16.6%) gray, 53 (11.3%) chestnut, 243 (51.6%) bay, and 97 (20.5%) black horses.

### 2.2. Study Design

The data for this study was obtained from the animals at rest in the boxes provided for them at the equestrian centers of the show jumping competition they were participating in. The measurements were taken one day before the official competition was held. The temperature in the boxes ranged from 14 °C to 24 °C and the relative humidity was between 40% and 50%. However, the owners were asked to arrive at the equestrian center between 12 and 24 h before measurement time, so that the horses could adapt to the new environmental conditions at the center before the measurements were taken. Thus, the horses were at rest at least 36 h before the official show jumping competitions. During their stay, the animals were housed in 3 × 3 m^2^ stall boxes with dry straw as bedding material and were fed with hay, concentrate, and water ad libitum, thus providing standardized environmental and housing conditions for all the animals. Measurements were taken in the morning, between 8:00 a.m. and 12:00 p.m. for all the horses, in 3 different geographical areas of Spain, depending on the locations where the equestrian competitions were held, including: central Spain (128 horses; 4 different events), eastern Spain (221 horses; 6 events) and southern Spain (122 horses; 4 events). All equestrian events were held during the same season (Spring: March, April, and May) in two consecutive years, 2021 and 2022, with temperatures ranging from 10 °C to 28 °C and humidity ranging from 35% to 60%. Events programming was designed to avoid overlapping between competitions. Furthermore, the ranking of these horses in the show jumping competitions held the day after the study was collected and recorded as values ranging from 1 (first and best ranking position) to 72 (last and worst ranking position).

### 2.3. Physiological Data

The physiological changes undergone by the animals at rest in the stall boxes were assessed with eye temperature (ET) measurements. To take the infrared photographs, all the horses were handled by their owners or their regular horse keeper inside their stall box. The camera operator stood 0.5–1 m away from the horse, perpendicular to its left eye, where the images were taken without touching the horse. To get the horses used to the operators and to the camera itself, a short period of habituation was performed 12 h before the test day with all the horses. During this habituation period, the horse could freely sniff the camera and the camera operator. It must be mention that most horses were already familiar with the camera operator from previous and regular sport competitions, as the researches from the study were also the responsibility of the management of the CDE’s Breeding Program. Thus, no long habituation periods were required.

Eye temperature images were always taken by the same person, with a portable infrared thermography camera (FLIR E90. FLIR Systems AB, Danderyd, Sweden). The emissivity was set to 0.98. In order to calibrate the camera results, environmental temperature and relative humidity were recorded with a digital thermo-hygrometer (Extech 44550, Extech Instruments, Nashua, New Hampshire) every time an eye temperature sample was taken. To determine eye temperature, FLIR Tools 6.0.17046.1002 (FLIR Systems AB, Danderyd, Sweden) image analysis software was used, measuring the temperature (°C) within an oval area traced around the caruncle of the eye, including approximately 1 cm around it. The program provided the maximum, minimum, and mean temperature of this oval area, but for the study purposes, only the maximum temperature was used (Figure 1). Three to four images were taken per animal. Later, the best image was selected to obtain the data for the study, with eye temperature at rest (ETR) being the variable analyzed in this study.

### 2.4. Statistical and Genetic Analyses

A previous Shapiro–Wilk test (results not shown) presented a normal distribution of the ETR variable. Hence, parametric statistical analyses were used.

To test the influence of horse characteristics on the physiological reactivity of the horses studied, a univariate one-way general linear model (GLM) analysis was performed for the ETR variable measured, considering the following 4 effects: age, with 3 levels: ≤4 years old (10.6%), 5 to 10 years old (13.2%), and ≥11 years old (76.2%); sex, with 2 categories: male (66.0%) and female (34.0%); location, with three categories: central Spain (27.2%), eastern Spain (46.9%), and southern Spain (25.9%); and breed group (as portrayed in Table 1).

To analyze the differences in the analyzed variable due to these effects, a least square means analysis with a Duncan post-hoc test was computed.

Furthermore, to check the relation between the final classification obtained by the animals measured in the equestrian competition held the day after the study (RANK), the ETR assessed in all animals and the coat color, a Spearman rank analysis was performed.

For the statistical analyses, the ‘Statistica for Windows’ software v. 12. (Stat Soft. Inc., Tulsa, OK, USA) was used.

The genetic parameters for the ETR variable were estimated. To do this, pedigree information was gathered from the studbook provided by the National Association of Spanish Sport Horse Breeders (ANCADES). The pedigree was traced back to 4 generations, with a total of 8245 animals (3146 males and 5099 females).

A BLUP genetic evaluation was computed based on a univariate animal model, with ETR as the variable. For the analyses, two genetic models were applied and compared: one model with genetic groups (MGG) and one with heterogeneous residual variance (MHRV). Both were performed using a Bayesian approach via Gibbs sampling using the GIBBSF90+ module of the BLUPF90 software [42]. The Gibbs sampler was run for 100,000 rounds, with the first 10,000 considered as burn-in and then every 10th sample saved for later analysis. Posterior means and standard deviations were calculated to obtain estimates of variance components. Convergence of the posterior parameters was assessed by visual inspection of trace plots of posterior distributions generated by the Coda R package [43].

The equation in matrix notation to solve the mixed model was:y = Xb + Zu + e,
$$\mathrm{with}\;(\begin{array}{c} u\\ e \end{array})\sim N([\begin{array}{c} 0\\ 0 \end{array}],[\begin{array}{cc} A\sigma_{u}^{2} & 0\\ 0 & I\sigma_{e}^{2} \end{array}]),$$
where y is the vector of observations, b the vector of systematic effects, u the vector of direct animal genetic effects and e the vector of residuals. X and Z are incidence matrices of systematic and animal genetic effects, respectively. $\sigma_{u}^{2}$ is the direct genetic variance, $\sigma_{e}^{2}$ the residual variance, I an identity matrix, and *A* the numerator relationship matrix.

More specifically, for the genetic groups model, b included age (3 levels: <5 years old; 5 to 10 years old; >10 years old), sex within coat color (8 levels), location (3 levels: central, Eastern and southern Spain), breed group (5 levels; Table 1), the animal as a random effect, and 5 genetic groups according to the coat color of the ancestor (chestnut, bay, black, gray, and unknown). In contrast, for the residual variance model, b included age (3 levels: <5 years old; 5 to 10 years old; >10 years old), sex (2 levels: male; female), coat color (4 levels: chestnut, bay, black, and gray), location (3 levels: central, eastern, and southern Spain) and breed group (5 levels; Table 1) as fixed effects.

To calculate the genetic evaluation, we estimated phenotypic variance, heritability of the animal, and the residual effect for the ET variable, for both genetic models. Finally, we computed estimated breeding values (EBV) for both genetic models and standardized them for an interval of 80–120, with a population average of 100.

### 2.5. Ethic Statement

No specific ethical approval was required for this study, since the infrared thermography measurements were obtained at rest. As regards the temperature samples obtained from the animals, since the eye temperature assessed with infrared thermography was a non-invasive physiological measure collected from a minimum distance of 0.5–1 m away from the animal, it was not necessary to obtain a specific permit for animal experimentation since the animals were not subjected to any pain or physical stress. Likewise, all the owners of the animals had been previously informed of the entire procedure and the type of samples to be taken, and the approval of all the owners of the animals participating in the study had been obtained.

## 3. Results

### 3.1. Statistical Analyses

In order to study the influence of environmental effects on the basal reactivity of an animal, a univariate one-way GLM variance analysis was performed including age, coat color, sex, location, breed group, sex within coat color, age within coat color, breed group within coat color, and location within coat color as fixed effects, and ETR as the variable analyzed (Table 2).

To study the differences identified by the GLM analysis, a least square means analysis with a Duncan post-hoc test was computed for the statistically significant effects found in Figure 2.

The results indicated that first, only six effects from the nine analyzed were statistically significant (*p* < 0.05) in the GLM analysis. Of these, gray horses had the lowest ETR values, whereas the youngest horses (≤4 years old) had the highest. Conversely, female horses generally demonstrated higher ETR values. Additionally, when the interaction with coat color was analyzed, female chestnut, black, and bay horses and male black horses portrayed higher ETR than male chestnut, bay, and gray horses. With respect to the location where data was collected, horses had the highest ETR values in competitions from central and eastern Spain. Finally, horses from BG4 had the lowest ETR results, with no differences between the rest of the breed groups.

Spearman rank correlations were computed between ETR values and rank level for the animals analyzed, obtaining a statistically significant (*p* < 0.05) positive correlation of 0.11. When the correlation was assessed according to the horse’s coat color, only statistically significant (*p* < 0.05) correlations were discovered for black horses, with a value of 0.23. Our results indicated a positive relation with ETR and the ranking position achieved in the sport competition for all the animals in general, although this was more evident in black horses. Thus, it followed that the higher the ETR, the worse the final classification of the animal in the show jumping competitions.

### 3.2. Genetic Parameters

Genetic parameters for the physiological variable ETR were computed with two different genetic models, using five genetic groups according to coat color and obtaining a genetic estimation for every coat color, considering residual variance heterogeneity (Table 3).

These results indicated genetic variability in the residual variance of ETR variable due to the horse’s coat color, with the gray coat indicating the highest heritability (0.22) and the black coat showing the lowest (0.17). Conversely, the model with genetic groups demonstrated an intermediate heritability value of 0.20.

Finally, in order to assess the differences between the estimations of the two genetic model, Figure 3 portrays a comparison of EBVs calculated with both genetic models, representing 20% of the animals from the genetic evaluation, with their best EBVs (1649 animals), distributed by coat color group, and indicating the number of coincident animals between methodologies for each group.

The results indicated that the highest EBVs were achieved for chestnut-coated horses estimated with the MGG model (111.9), followed by black-coated horses estimated with the MHRV model (111.8). In contrast, the lowest EBV estimation was obtained for gray-coated horses with the MGG model. With respect to the number of coincident animals between the genetic models, the black- and chestnut-coated horses indicated the highest percentage of coincidence, with 98.2% and 98.1%, respectively, whereas the gray- and bay-coated horses had the lowest (76.9% and 75.2%), respectively. Generally, EBVs were higher using the MHRV model than the MGG, thus indicating a higher environmental variance of the ETR due to coat color.

## 4. Discussion

Behavior, reactivity, and temperament are valuable quantitative traits in all domesticated livestock species. No production system can operate effectively when excessive fear, aggression, or abnormal behavior jeopardizes the safety and well-being of the animals and their human caretakers [19]. This is especially true in horses, because of their value as a performance animal and consequent close interactions with humans. Previous studies have reported different environmental effects that could determine considerably the physiological response of a horse to certain circumstances (sex, rider, weather, etc.), due to their reactivity and temperament [16,19,44,45]. Some of these factors are genetically determined, such as sex, breed, or coat color. Thus, knowing the size of this reactivity and expression of stress, which is already programmed in the horse from birth could be a useful tool for horse breeding programs in general, and for the CDE breeding program in particular, as it could help them plan horse mating to obtain animals that could adapt better to the new stimuli found in the sport competitions they are trained for.

The pleiotropic effects of coat color associated with mutations in the same genes have been widely reported in horses in relation to different pathologies, like osteochondrosis [46] or melanoma [47] related with grey coat color genes (MC1R), as the negative influence that has a certain frequency of coat color genes on the fertility parameters of a breed [48] and also related to different behavior and reactivity patterns, like more independency of ASIP gene coats [14], abnormal behaviors and cautious temperament related to silver coat color [15,16] or a bold character of chestnut-coated horses [18]. These coat color differences, related to reactivity, lead to differences in perceiving new stimuli. As the parameter measured in this study was eye temperature at rest in horses with different coat colors, this physiological response (detected by the infrared thermography as an increase in eye temperature) is probably related with the animal’s innate tendency to react to new stimuli, which is termed reactivity [32,33,35,44].

First, with respect to GLM and the post-hoc results, Gray-coated horses portrayed the lowest ETR values, compared with chestnut-, bay- or black-coated horses. Bartolomé et al. [49], found differences in the CDE population structure due to coat color, with gray horses presenting a slight genetic differentiation from other coat colors. Furthermore, different authors [14,15,16] have also reported differences in coat colors related to personality traits such as calmness, boldness, or nervousness which, to some extent, are related with the way the animal perceives threat stimuli and reacts by showing stress [16,50,51,52,53]. Conversely, some studies developed in humans found a lower pain threshold in red-haired people. Genetic variations in the MC1R gene have been identified related to alterations in pain perception and sensitivity to analgesics [14,54,55]. Although the pharmacodynamics involved in analgesics are complex, coat color in the horse could play an important role as a possible phenotypic indicator of variations in sensitivity to opioids [55] and thus, in the response to stressors.

Results discovered for sex and age were in accordance with previous studies on stress in horses assessed with infrared thermography [33,36,38], confirming that young horses and females tend to show higher stress values at rest (ETR), due to their higher basal reactivity levels. Burattini et al. [56] also reported differences in behavior patterns due to age of the horse, which agrees with our results, as older horses were bolder than younger horses. However, Aune et al. [57] reported the opposite results to ours with respect to the influence of sex on horse behavior, as they discovered that behavior differences due to sex in sport horses were more closely related with riders’ and breeders’ pre-conceptions than to real differences between them. This could be because these authors were not measuring basal reactivity, but different fear, stereotypic, and aggressive behaviors under the rider, thus assessing more horse–rider interaction than reactive behavior in horses.

As regards sex–coat color interaction, chestnut and bay female horses showed the highest ETR differences, with male horses having the same coat colors. These results were in line with previous studies which linked bay coat colors with more nervous characters, not only in horses [14], but also in foxes [58], rats [59], or dogs [50] more reactive animals [15]. Otherwise, another study [14] discovered no correlation between the chestnut genotype and the behavior test they performed, while [18] found no differences in behavior patterns between chestnut and bay horses. Additionally, Corbin et al. [60], in a genome-wide association study (GWAS), discovered pleiotropic effects between bay-coated horses, behavior, and obesity.

Results demonstrating that competitions held in southern Spain were linked to horses with the lowest ETR values could be because CDE horses competing in this area tend to be calmer and less reactive to their environment. These results could also bear a relationship to the breed group results, as the PRE breed group also demonstrated the lowest ETR values and, since this breed is mainly bred in southern Spain [61], most of the animals participating in these competitions could be CDE horses from the PRE breed group, as these horses tend to be less reactive and bolder than horses from other breeds [38,62]. Thus, the physiological response of these CDE crossbred animals at rest measured one day before the equestrian competition seems to be conditioned by the physiological response of the breeds included in their pedigree.

Correlations discovered between ETR and RANK results in the show jumping competition held after the sample collection corroborated previous studies in horses indicating that basal reactivity and tendency to get easily stressed by novel stimuli affects the horse’s sport performance, giving them a lower chance of reaching the top positions in the classification [26,27,28,30,33,36]. Furthermore, this correlation increased with black-coated horses, indicating that there is a greater influence of reactivity changes in these horses’ performance. However, Stachurska et al. [12] and Negro et al. [62] accounted for this relationship by highlighting possible close gene links or pleiotropy in certain genes.

With respect to the genetic models used for the genetic evaluation, with the MGG, a homogeneous residual variance for any given genotype is assumed [63,64] and thus, the coat color effect is assumed as a genetic effect with no (or very little) environmental variation. Conversely, the MHRV models assume that genotypes at individual loci differ not only in their effect on the mean but also on the variance of the trait evaluated [65,66]. In our study, differences in basal reactivity in horses due to coat color could have both a genetic and a residual origin. Thus, it would be useful to compare these models to assess the main origin of the coat color differences in CDE horses, in order to select them by their basal reactivity.

Estimates of h2 for residual variance (0.17–0.22) were similar to those reported by Domínguez-Viveros [67] in behavior traits of fighting bulls (0.07–0.22), but lower than those reported by Mulder et al. [68] for live weight in birds (0.23–0.34) or by Fina et al. [69], evaluating birth weight in Pirineus cattle (0.44). Conversely, MGG results were higher than those reported previously in sport horses due to breed origin [70].

Furthermore, when EBVs were computed with both genetic models, the generally higher EBVs computed with the MHRV indicated that part of the variability present in the basal reactivity because coat color has an environmental origin. Thus, when the genetic models used for the genetic parameters take that factor into consideration (using MHRVs instead of MGGs), the data fit better and the EBVs are higher. These results are aligned with previous studies comparing different genetic models for animal production [70,71,72]. Only black-coated horses obtained higher EBVs with GGM than with MHRV. This could be because this coat is frequently associated with more self-reliant animals [14], and with consistent behavior regardless of the environment. Conversely, this response could be also related to a different dynamic in thermoregulatory responses demonstrated by animals with dark coats compared with those with lighter coats [73], which could have affected the ETR measurement evaluated in this study.

These coat color differences are of crucial concern in selection plans for a composite breed without morphological/phaneroptic standards like the CDE, since they could be used to offer different breeding recommendations to CDE breeders according to the horse’s coat color. Thus, in further studies, it would be interesting to assess basal reactivity in horses before a sport competition together with other physiological and genomic tools, in order to corroborate the IRT results obtained in this study.

## 5. Conclusions

Basal reactivity, assessed in CDE sport horses before a show jumping competition using infrared thermography, is influenced by several environmental effects, such as coat color, sex, age, location of the competition and CDE breed group. A correlation with the ranking position in show jumping was also found, with less reactive horses being more likely to achieve better rankings. The genetic parameters estimated for the ETR variable were similar for both genetic models, with higher heritability values obtained with the MHRV. However, the genetic evaluation resulted in higher EBVs for the ETR when coat color was due to heterogeneous residual variance, thus indicating the pleiotropic effect of coat color, both with genetic and environmental effects. Black coat color was the only one demonstrating higher EBVs for the genetic groups model, probably due to a stronger genetic effect behind the IRT measured in this coat color than for the others. These results indicate that this parameter has a suitable genetic basis to be used in the CDE Breeding Program to select for basal reactivity due to coat color.

## Figures and Tables

**Figure 1 animals-12-02515-f001:**
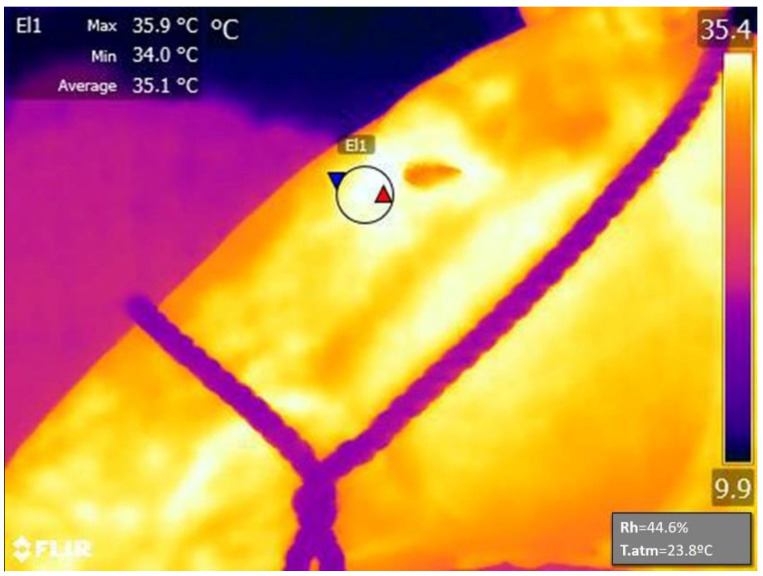
Infrared thermography image of the eye region. The image was taken at rest, before an equestrian competition. El1 indicates the area of the photo where the measurements were taken (located at the medial posterior palpebral border of the lower eyelid and the lachrymal caruncle). Square up and left of the image indicate maximum (Max; 
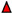
), minimum (Min; 
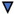
) and average temperature (°C) registered within the El1 area. Square down and right of the image indicate the relative humidity (Rh) and the atmospheric temperature (T.atm) registered during the measurement and used to calibrate the results. The color scale on the right indicated maximum and minimum temperatures registered within the photo.

**Figure 2 animals-12-02515-f002:**
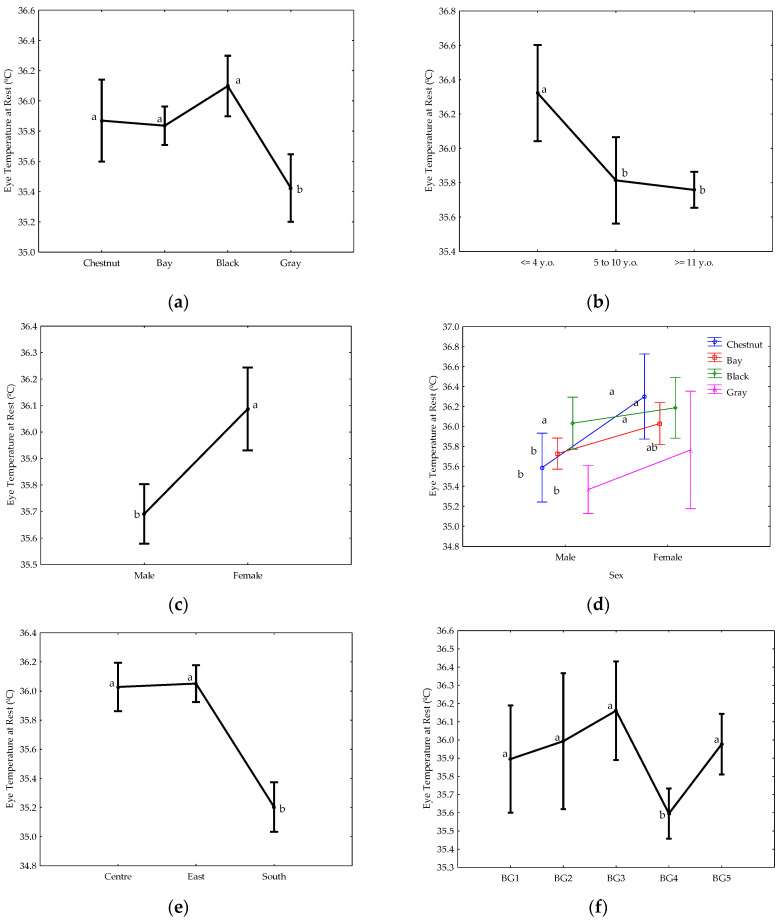
Least square means analysis with a Duncan post-hoc test for the coat color (**a**), age (**b**), sex (**c**), sex within coat color (**d**), location (**e**) and breed group (**f**). Different letters indicate statistically significant differences between levels (*p* < 0.05). Center: central Spain; east: eastern Spain; south: southern Spain. Breed lines codes (BG1 to BG5) are indicated in Table 1.

**Figure 3 animals-12-02515-f003:**
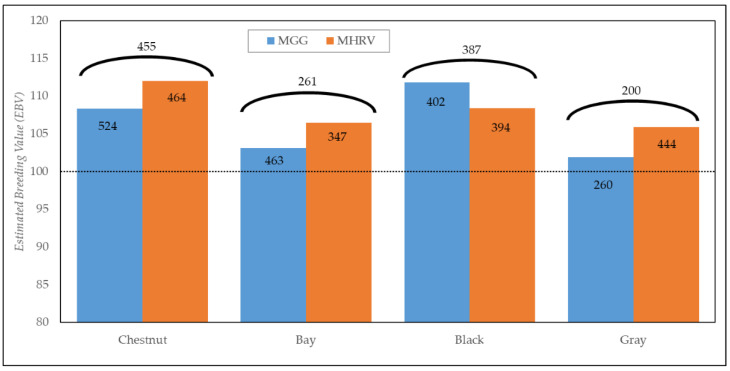
Estimated breeding values (EBV) calculated with a model based on genetic groups (MGG) and a model based on variance heterogeneity (MHRV), for 20% of the animals genetically evaluated, with their best EBVs, according to coat color groups. The numbers in the columns indicate the number of animals in each group, while the numbers above the columns indicate the number of coincident animals between the genetic models. The black dotted line marks the mean EBV (100).

**Table 1 animals-12-02515-t001:** Description of the breed groups of the Caballo de Deporte Español animals measured.

Breed Group	Name	Description	N (%)
BG1	German (GE)	More than 50% of the CDE ancestors belonged to German horse breeds: Holsteiner, Hanoverian, Westphalian, Oldenburger, or Trakehner.	45 (9.6%)
BG2	Netherlands (NH)	More than 50% of the CDE ancestors belonged to Netherland horse breeds: Dutch Warmblood, Belgian Warmblood.	28 (5.9%)
BG3	Trotter (TR)	More than 50% of the CDE ancestors belonged to Trotter horse breeds.	141 (29.9%)
BG4	Pura Raza Español (PRE)	More than 50% of the CDE ancestors belonged to Pura Raza Español breed.	204 (43.3%)
BG5	Other Horse Breeds (OHT)	Included CDE horses with more than 50% of their ancestors from other minority sport horse breeds (Zangersheide, Irish Sport Horse, etc.).	53 (11.3%)

**Table 2 animals-12-02515-t002:** General linear model analysis for the eye temperature at rest (ETR) variable, considering different fixed effects.

*p* Value		Effect			
		Age	Sex	Location	Breed Group
Total Population		**	***	***	***
Coat color ***	Total Interaction	n.s.	**	n.s.	n.s.
	Chestnut	n.s.	**	**	n.s.
	Bay	n.s.	*	n.s.	n.s.
	Black	*	n.s.	n.s.	n.s.
	Gray	n.s.	n.s.	n.s.	**

n.s.: not statistically significant, * *p* < 0.05, ** *p* < 0.01, *** *p* < 0.001.

**Table 3 animals-12-02515-t003:** Genetic parameters for eye temperature at rest according to coat color, calculated with different genetic models.

Model		σ_u_			σ_e_			h^2^
		Mean (s.d.)	Median	HPD 95%	Mean (s.d.)	Median	HPD 95%	
MGG		0.186 (0.145)	0.149	0.007–0.476	0.736 (0.144)	0.760	0.425–0.977	0.20
MHRV	Chestnut	0.201 (0.092)	0.190	0.056–0.393	0.82 (0.221)	0.799	0.358–1.2340	0.20
	Bay				0.88 (0.125)	0.883	0.665–1.141	0.19
	Black				0.98 (0.175)	0.973	0.667–1.319	0.17
	Gray				0.72 (0.180)	0.706	0.394–1.1020	0.22

MGG: model with genetic groups; MHRV: model with heterogeneous residual variance; σ_u_: additive genetic variances; σ_e_: residual variances; h^2^: heritability; s.d.: standard deviations; HPD 95%: highest posterior density (95%).

## Data Availability

The data presented in this study are available on request from the corresponding author. The data are not publicly available due to permission from the Breeder’s Association is needed.

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
