# Peer review of "Basal Reactivity Evaluated by Infrared Thermography in the “Caballo de Deporte Español” Horse Breed According to Its Coat Color"

_animals, 2022, doi:10.3390/ani12192515_

Round 1

Reviewer 1 Report

Line 12: What is ETR? basal reactivity or infrared thermography or eye temperature at rest? currently, there is a contrast between lines 12, 24, and 146. Please check and correct accordingly.

Line 26: Full version of GLM should be used here as this is the first time it appears in the text. Please edit it.

Line 50-56: The physiological and biochemical changes related to stress was mentioned here. I would suggest the inclusion of behavioral changes related to stress as well. For author’s help, https://www.mdpi.com/2076-2615/11/5/1333 ; https://www.frontiersin.org/articles/10.3389/fpsyg.2019.00849/full

Line 68-69: “As occurs with heart rate or lactate levels, eye temperature assessed with IRT is not 68 only dependent on the aerobic capacity of the horse” Please provide a reference for this line.

Line 77-78: Word 'different' is vague. Please enlist the environmental effects that you have included in your study.

Line 78-79: To make this sentence a bit clear, please replace 'assessed with IRT' with 'using IRT'.

Line 98-99: Please rewrite this sentence, it is unclear at the moment. My suggestion is,

"When the majority of an animal's ancestors came from a breed in one of these groups, it was classified as belonging to that group."

Line 113-116: Can horses adapt to a new environment in 12 hours? If so, please provide a reference. Please rewrite this sentence as well as it is hard to read at the moment. I would suggest making two clear sentences of one very large sentence.

Line 119-122: How many groups of researchers were involved in taking measurements in four hours, between 08AM to 12PM.? As you had 471 horses in three different regions. Please clarify. If there were more than one team involved or study period was different for each region. If so, then how this will impact overall study results? Please edit accordingly.

Line 130-132: What was the distance between eye and camera? 0.5m or 1m or it varied between 0.5-1m? As there is a contrast between lines 131 and 204. Please clarify and make changes accordingly.

Line132-135: How 471 horses were habituated with IRT camera and operators in 12 hours. I think detail should be mentioned regarding this in the study design. Please see my previous comment on this.

Line 145-147: Could you please provide an image showing the measurement of eye temperature with FLIR Tools software in the manuscript?

Line 201: “No specific ethical approval was required for this study”

I think this study involves live and sentience animals along with the introduction of equipment and persons unknown to horses. So, it was necessary to have ethical oversight.

Line 299: Please provide names of pathologies here. Currently word 'different pathologies; is vague.

Line 300: What are those behavior and reactivity patterns.? Please add this in the text.

Author Response

Line 12: What is ETR? basal reactivity or infrared thermography or eye temperature at rest? currently, there is a contrast between lines12, 24, and 146. Please check and correct accordingly.

ANS: ETR is eye temperature at rest. We corrected the text accordingly (L11; L24; L161).

Line 26: Full version of GLM should be used here as this is the first time it appears in the text. Please edit it.

ANS: Word edited (L27).

Line 50-56: The physiological and biochemical changes related to stress was mentioned here. I would suggest the inclusion of behavioral changes related to stress as well. For author’s help, https://www.mdpi.com/2076-2615/11/5/1333 ;https://www.frontiersin.org/articles/10.3389/fpsyg.2019.00849/full

ANS: References were included and the paragraph was rephrased (L55-56).

Line 68-69: “As occurs with heart rate or lactate levels, eye temperature assessed with IRT is not only dependent on the aerobic capacity of the horse” Please provide a reference for this line.

ANS: References were included accordingly (L76).

Line 77-78: Word 'different' is vague. Please enlist the environmental effects that you have included in your study.

ANS: The environmental effects studied were included in the text accordingly (L85).

Line 78-79: To make this sentence a bit clear, please replace' assessed with IRT' with 'using IRT'.

ANS: Expression replaced (L86).

Line 98-99: Please rewrite this sentence, it is unclear at the moment. My suggestion is, "When the majority of an animal's ancestors came from a breed in one of these groups, it was classified as belonging to that group."

ANS: Sentence was rephrased (L106-107)

Line 113-116. Can horses adapt to a new environment in 12 hours? If so, please provide a reference. Please rewrite this sentence as well as it is hard to read at the moment. I would suggest making two clear sentences of one very large sentence.

ANS: Despite new environment adaptation is difficult to assess, previous studies have reported an adequate recovery from sport performance from 120 minutes to 3 hours (Bitschnau, C., T. et al. 2010. Equine Vet. J. Suppl. 42:17–22; Bartolomé, E., et al. 2021. Animals. 11:1–11; Gordon, M. E., K. H. et al. 2007. Vet. J. 173:532–540). Taking into account that physiological parameters altered in sport performance (heart rate, lactate, cortisol, eye temperature, etc.) are similar to those altered in stress produced by a new environment (heart rate, cortisol, eye temperature, adrenaline, etc.), 12h could be considered enough. On the other hand, most horses stayed longer, as 12 hours was the minimum time required to participate in this study. Most horses stayed between 12 and 24h at the event facilities before the study. Despite we would have prefer to obtain data from animals that were at least 24h or more before the measurements at rest were taken, the reviewer has to consider that measurements were taken in regular competitions, where the owners kindly agreed to participate in our study with their horses. Participating in our study implied that they had to arrive earlier to the event center, having to afford also all the extra-cost that it would imply renting the box for their horses for more time. That is why we did not ask for a longer “previous minimum time” to be at the facilities. Furthermore, we also considered 12h as an adequate “minimum time” due to all the horses already had experience of participating in previous competitions held at the same locations. According to your suggestions, sentence was rephrased for a better understanding (L121-123).

Line 119-122: How many groups of researchers were involved intaking measurements in four hours, between 08AM to 12PM.? As you had 471 horses in three different regions. Please clarify. If there were more than one team involved or study period was different foreach region. If so, then how this will impact overall study results? Please edit accordingly.

ANS: The competitions were held at different days within and between regions and the programming was designed to avoid overlapping between competitions. All horses were measured by a unique research team, measuring all horses within the same season (Spring), in two consecutive years (2021 and 2022) in a total of 14 equestrian competitions held between March and May, 2021 and 2022. The year was analyzed in the GLM model but appeared as not statistically significant, so, it was not considered for the final model used in the study. More details were included in the text accordingly (L129-134).

Line 130-132: What was the distance between eye and camera? 0.5m or 1m or it varied between 0.5-1m? As there is a contrast between lines 131 and 204. Please clarify and make changes accordingly.

ANS: The camera was held 0.5m to 1m away from the eye of the horse. Text was corrected accordingly (L142).

Line132-135: How 471 horses were habituated with IRT camera and operators in 12 hours. I think detail should be mentioned regarding this in the study design. Please see my previous comment on this.

ANS: As clarified previously, horses had to be at the equestrian center between 12 and 24h before the test day. So that, the habituation with camera and researchers could be made when they arrived, at least 12 hours before the competition day for all the horses. Furthermore, most horses were already familiar with camera operators due to they were the researchers in charge of the CDE Breeding Program. Thus, they use to assist to most competitions and particular studs, due to the CDE Breeding Program management. Text was changed accordingly (L146-149).

Line 145-147: Could you please provide an image showing the measurement of eye temperature with FLIR Tools software in the manuscript?

ANS: A new figure 1 was included and the rest of the figures were renumbered accordingly (L159; L163-170).

Line 201: “No specific ethical approval was required for this study”. I think this study involves live and sentience animals along with the introduction of equipment and persons unknown to horses. So, it was necessary to have ethical oversight.

ANS: As indicated previously, researchers were not completely unknown to horses due to they are regularly present in this type of competitions, participating in the collection of data from the animals (competition results, morphological evaluations, behavior tests, etc.) for the CDE Breeding Program. Furthermore, no specific ethical approval was needed for similar previous studies developed by our research team with horses, already published in Valera, M., et al. 2012. J. Equine Vet. Sci. 32:827–830.; Bartolomé, E., et al. 2013. Animal, 7(12), 2044-2053; Sánchez, M. J., et al. 2016. Appl. Anim. Behav. Sci. 174 or Negro, S., et al. 2018. Res. Vet. Sci. 118. On the other hand, each horse’s owner was informed with all the characteristics of the study and they all gave their consent to develop the research with their animals. For all this, we considered that no further permissions would be needed here.

Line 299: Please provide names of pathologies here. Currently word' different pathologies; is vague.

ANS: The names of the pathologies was provided as suggested (L322-323)

Line 300: What are those behavior and reactivity patterns.? Please add this in the text.

ANS: Behavior and reactivity patterns were detailed as suggested (L325-327).

Reviewer 2 Report

The research for identification the Basal reactivity evaluated by infrared thermography in the “Caballo de Deporte Español” horse breed according to its coat color is of great interest in the breeders, trainers, owners and veterinarians. The present study has sought the key issues which such as the quality of horsemanship, horses’ health, educating horses to cope will provide the best potential to improve breeding systems and animals welfare by adequate horses maintenance connected with the temperament. However, the paper needs several correction.

1.      First of all, The Authors should add the information why the coat color is interesting for the breeders. Is it connected with the performance? Sometimes it is important  for example in endurance horses because grey color is more desirable.

2.     Also some information if the color is connected with some diseases should be added.

3.     Materials and methods – the huge strength of the research is the number of horses however the limitation part connected with weather conditions should be added.

4.     In discussion part should be added which specific genes are connected with behavioural reactions. Limitation of the study is that genome research was not performed.

5.     In my opinion other factors such as cortisol measurements will be improving the study quality however I understand that it is not always possible to perform blood sampling in clinical studies.

6.     Also some studies performed in humans confirm that red color hair people have got lover pain threshold. Mabey red horses reactivity is connected also with different receptors reaction as well.

Author Response

The research for identification the Basal reactivity evaluated by infrared thermography in the “Caballo de Deporte Español” horse breed according to its coat color is of great interest in the breeders, trainers, owners and veterinarians. The present study has sought the key issues which such as the quality of horsemanship, horses’ health, educating horses to cope will provide the best potential to improve breeding systems and animals welfare by adequate horses maintenance connected with the temperament. However, the paper needs several correction.

  1. First of all, The Authors should add the information why the coat color is interesting for the breeders. Is it connected with the performance? Sometimes it is important, for example in endurance horses because grey color is more desirable.

ANS: The coat color is certainly a very interesting character, sought by the breeders due to the pleiotropic effect it has with physiological, health, behavioral or even morphological traits. And also, simply due to breeders' own preferences and previous experience or pre-conceptions about its relation with other characteristics in the animal. For example, if they are looking for calm, concentrated and easy-to-handle horses, they would probably prefer gray coated horses. Whereas, if they are seeking for more active/temperamental animals that would perform better in equestrian competitions, other coat colors as black or chestnut would be probable more desirable for breeders. As regards to the connection of coat color with performance, despite most papers have declined a direct relation (Stachurska, A., et al. 2007. Livest. Sci. 106:282–286.; de Mira, M. C., et al. 2021. BMC Vet. Res. 17:1–12; Junqueira, G. S. B., et al. 2021. J. Appl. Genet. 62:297–306.), some pleiotropic effects with different factors influencing performance were found (Ripollés-Lobo, M., et al. 2022. Livest. Sci., 105031; Reissmann, M., and A. Ludwig. 2013. Semin. Cell Dev. Biol. 24:576–586.; Brunberg, E., et al. 2013. Appl. Anim. Behav. Sci. 146:72–78.; Poyato-Bonilla, J., et al. 2018. Ann. Anim. Sci. 18:723–739). This would have probably influence indirectly the breeder’s ancient belief about this relation.

According to Reviewer 1 suggestions, some examples of the pleiotropic effects related with coat color that affects performance, behavior, physiological and health parameters are included in the Introduction (L43-48) and Discussion (L321-327) sections. All of them are of a great interest for horse breeders.

  1. Also some information if the color is connected with some diseases should be added.

ANS: This connection is already added in Discussion (L322-324). Some more details were also included in Introduction (L43-44).

  1. Materials and methods – the huge strength of the research is the number of horses however the limitation part connected with weather conditions should be added.

ANS: According to Reviewer 1 suggestions, some more explanations about the collection of the data was provided (L129-134). In Spain the climatic conditions are very favorable in the months considered in this study (March, April and May), throughout all the country. Temperature ranged from 10ºC to 28ºC and humidity rate, from 35% to 60% and both measures were used to calibrate infrared thermography camera. So, eye temperature measurements were already corrected by weather conditions. Furthermore, in a previous GLM analysis, we included the month and the year of measurement in the model as fixed effects (in order to account for the weather), but they resulted not statistically significant (p>0.05), so we did not include them in the final model.

  1. In discussion part should be added which specific genes are connected with behavioral reactions. Limitation of the study is that genome research was not performed.

ANS: Some genes specifications were included (L325-327) and limitations were added (L414).

  1. In my opinion other factors such as cortisol measurements will be improving the study quality however I understand that it is not always possible toper form blood sampling in clinical studies.

ANS: Due to the animals evaluated were participating in regular equestrian competitions, it was not possible for us to obtain any blood sample from horses. Likewise, we have previously analyzed salivary cortisol in these type of animals and compared it to infrared thermography results (Valera, M., et al. 2012. J. Equine Vet. Sci. 32). However, as no clear relations were found, we decided to use other physiological parameters instead.

  1. Also some studies performed in humans confirm that red color hair people have got lover pain threshold. Mabey red horses reactivity is connected also with different receptors reaction as well.

ANS: According to your suggestions, these references were included in the Discussion section (L339-345).